# A Clinical Study of Alveolar Bone Tissue Engineering Using Autologous Bone Marrow Stromal Cells: Effect of Optimized Cell-Processing Protocol on Efficacy

**DOI:** 10.3390/jcm11247328

**Published:** 2022-12-09

**Authors:** Hideaki Kagami, Minoru Inoue, Hideki Agata, Izumi Asahina, Tokiko Nagamura-Inoue, Masataka Taguri, Arinobu Tojo

**Affiliations:** 1Division of Stem Cell Engineering, The Institute of Medical Science, The University of Tokyo, Tokyo 108-8639, Japan; 2Tissue Engineering Research Group, Division of Molecular Therapy, The Advanced Clinical Research Center, The Institute of Medical Science, The University of Tokyo, Tokyo 108-8639, Japan; 3Department of Oral and Maxillofacial Surgery, Matsumoto Dental University, Shiojiri 399-0781, Japan; 4Institute for Oral Science, Matsumoto Dental University, Shiojiri 399-0781, Japan; 5Department of Dentistry and Oral Surgery, Aichi Medical University, Nagakute 480-1195, Japan; 6Inoue Dental Clinic, Shizuoka 420-0866, Japan; 7Agata Dental Clinic, Hamamatsu 430-0929, Japan; 8Unit of Translational Medicine, Department of Regenerative Oral Surgery, Graduate School of Biomedical Sciences, Nagasaki University, Nagasaki 852-8523, Japan; 9Department of Oral and Maxillofacial Surgery, Juntendo University Hospital, Tokyo 113-8431, Japan; 10Department of Cell Processing and Transfusion, The Institute of Medical Science, Research Hospital, The University of Tokyo, Tokyo 108-8639, Japan; 11Division of Medical Data Science, Tokyo Medical University, Tokyo 160-8402, Japan; 12Institute of Innovation Advancement, Tokyo Medical and Dental University, Tokyo 113-8510, Japan

**Keywords:** bone tissue engineering, bone marrow, mesenchymal stromal cells, mesenchymal stem cells, clinical study, atrophic alveolar bone

## Abstract

(1) Objectives: The effect of cell-processing protocols on the clinical efficacy of bone tissue engineering is not well-known. To maximize efficacy, we optimized the cell-processing protocol for bone-marrow-derived mesenchymal stromal cells for bone tissue engineering. In this study, the efficacy of bone tissue engineering using this modified protocol was compared to that of the original protocol. (2) Materials and Methods: This single-arm clinical study included 15 patients. Cells were obtained from bone marrow aspirates and expanded in culture flasks containing basic fibroblast growth factor. The cells were seeded onto β-tricalcium phosphate granules and induced into osteogenic cells for two weeks. Then, the cell–scaffold composites were transplanted into patients with severe atrophic alveolar bone. Radiographic evaluations and bone biopsies were performed. The results were compared with those of a previous clinical study that used the original protocol. (3) Results: Panoramic X-ray and computed tomography showed bone regeneration at the transplantation site in all cases. The average bone area in the biopsy samples at 4 months was 44.0%, which was comparable to that in a previous clinical study at 6 months (41.9%) but with much less deviation. No side effects related to cell transplantation were observed. In regenerated bone, 100% of the implants were integrated. (4) Conclusions: Compared to the original protocol, the non-inferiority of this protocol was proven. The introduction of an optimized cell-processing protocol resulted in a comparable quality of regenerated bone, with less fluctuation. Optimized cell-processing protocols may contribute to stable bone regeneration.

## 1. Introduction

Patients who have lost their teeth due to severe periodontitis or who have been wearing dentures for a long period often experience alveolar bone atrophy. Bone regeneration therapy is mandatory for the installation of dental implants in severely atrophic alveolar bones. Autologous bone transplantation is the gold standard for bone regeneration therapies. However, harvesting autologous bone is inevitably accompanied by swelling and pain at healthy donor sites. Recently, artificial bones such as hydroxyapatite, β-tricalcium phosphate (β-TCP), and carbonate apatite have been widely used [1,2]. Most artificial bones only possess osteoconduction capability and lack osteo-induction and bone regeneration capabilities; thus, the size and shape of the target bone defect are limited. Accordingly, there has been a strong demand for novel transplantation materials that are not invasive, are more functional, and can be applied to severe alveolar bone atrophy cases.

Langer and Vacanti first introduced the concept of tissue engineering in 1993 [3]. This concept advocates three main elements for tissue regeneration: living cells, scaffolds, and growth factors. Bone marrow stromal cells (BMSCs) have been widely used in bone tissue engineering and have already been clinically applied [4,5]. Since 2004, our group has conducted a clinical study of alveolar bone regeneration using BMSCs. The transplants from all participants showed bone regeneration, and the integration of implants was achieved in 93% (27/29) [6], showing the feasibility of bone tissue engineering using BMSCs. However, non-negligible individual variation was noted in the histology of the regenerated bone, which might be due to the individual variation in BMSCs, such as cell proliferation capability and alkaline phosphatase (ALP) activity [7]. These results suggest the need for further optimization of cell-processing protocols and investigation of the effect of cultured cell quality on clinical efficacy.

Therefore, we optimized the cell culture and processing conditions of BMSCs for bone tissue engineering based on their ability to form bone in vivo [7] and generated a novel modified protocol for a clinical study on alveolar bone tissue engineering [8]. We report the results of a clinical study using this modified protocol and compared them with those of the previous clinical study [6].

## 2. Materials and Methods

This study conformed to the tenets of the Declaration of Helsinki, and the protocol was approved by the institutional review board of the Institute of Medical Science, The University of Tokyo (IMSUT) (IRB for clinical study using human stem cells, No. 21-1) and with the permission of the Minister of Health, Labour, and Welfare of Japan. All participants provided written informed consent. This study was registered in the University Hospital Medical Information Network (UMIN) Clinical Trials Registry (UMIN000006255).

The study design and time points for each intervention are summarized in Table 1.

### 2.1. Inclusion Criteria

The subjects should have continuous tooth defects (>2), where fixed prostheses are not applicable. The participants wished to undergo dental implant treatment rather than conventional removable prostheses. The subjects had severely atrophic maxillae or mandibles, which required bone transplantation. The width of the alveolar bone at the installation site was <5 mm. In the maxilla, the distance between the alveolar ridge and the sinus floor was <5 mm. Similarly, in the mandible, the distance between the ridge and the mandibular canal was <5 mm. Tooth-brushing and scaling were performed prior to protocol treatment, and good oral hygiene was maintained. The age of the participants was limited to 20–70 years.

### 2.2. Exclusion Criteria

Patients with diabetes and/or autoimmune diseases who presented hemorrhagic diathesis where partial thromboplastin time was <50% and activated partial thromboplastin time less than 23.5 s or longer than 42.5 s, who were taking anticoagulant or antiplatelet drugs of a type that were difficult to stop; who were positive for syphilis, anti-HBV antigen, anti-HCV antibody, anti-HTLV-1 antibody, or anti-HIV antibody; with osteoporosis, liver dysfunction with an aspartate aminotransferase value <10 or >40 IU/L or with an alanine aminotransferase <5 or >45 IU/L; who were pregnant or possibly pregnant; with an allergy for any of the medications used in this study and/or allergy that required continuous systemic medication, who smoked, or who had other special conditions that the responsible physician or dentist considered inappropriate, were excluded.

### 2.3. Number of Subjects and Duration of Study

The participants were referred from other hospitals/clinics and primarily fulfilled the eligibility criteria. No one was excluded after examination, and there was no drop out. Fifteen patients were enrolled in this study, and the follow-up period was 2 years after cell transplantation.

### 2.4. Donor Screening

Blood tests, urinalysis, chest X-ray, panoramic X-ray, computed tomography of the head, and electrocardiography were performed to screen the subjects before enrollment in this clinical study.

### 2.5. Autologous Serum Preparation

Peripheral blood and bone marrow were harvested, as previously described [6]. Briefly, prior to bone marrow aspiration, 200–400 mL of peripheral blood was collected and autologous serum was prepared. Peripheral blood was stored at 4 °C for 1 h and centrifuged. The serum was transferred to fresh bags (approximately 50 g each) for aseptic cryopreservation (Nipro Corp., Osaka, Japan). The bags were stored in a freezer at −20 °C until use. Before use, the serum was thawed and added to the culture medium containing antibiotics to a final serum concentration of 10%.

### 2.6. Harvest and Expansion of Bone Marrow Stromal Cells

Bone marrow was harvested from the iliac crest under local anesthesia. The marrow (10 mL) was aspirated into a syringe containing 500 U of heparin, and two syringes were used (total of 20 mL). The aspirated bone marrow was diluted four-fold with α-MEM (Gibco BRL, Grand Island, NY, USA) containing 10% autoserum, 2.5 µg/mL amphotericin B, and 50 µg/mL gentamicin sulfate. The medium containing cells was plated in 150-cm^2^ flasks (Corning, New York, NY, USA). On day 4, the culture medium was changed to medium containing basic fibroblast growth factor (bFGF; FIBLAST Spray 250, Kaken Pharmaceutical Co., Ltd., Tokyo, Japan), and non-adherent cells were discarded. Adherent cells were cultured as BMSCs [9]. The BMSCs were maintained in the same medium at 37 °C in a 5% CO_2_ atmosphere. The cells demonstrated a typical spindle shape that was maintained throughout the cell culture period. When the cells attained 80% confluence, they were subcultured into tubes containing 50 mg of β-TCP granules (OSferion, Olympus Terumo Biomaterials Corp., Tokyo, Japan). The next day, the medium was changed to osteogenic induction medium containing 10 nM dexamethasone (Dex, Sigma-Aldrich, St. Louis, MO, USA) and 100 μM ascorbic acid (Wako Pure Chemical Industries, Ltd., Osaka, Japan) in α-MEM with 10% autoserum, 2.5 µg/mL amphotericin B, and 50 µg/mL gentamicin sulfate and induced for 13–16 days before transplantation. The medium was changed twice a week.

#### 2.6.1. Test for Osteogenic Differentiation

Osteogenic differentiation of BMSCs was confirmed by ALP activity as previously described [10]. Briefly, 1 × 10^6^ of harvested cells were incubated with 400 µL of 20 mM HEPES (Dojindo, Kumamoto, Japan) buffer (pH 7.5) containing 1% Triton X-100 (Wako Pure Chemical Industries, Ltd., Osaka, Japan) to extract proteins. After extraction, the supernatants were incubated with BCA protein assay reagent (Pierce Thermo Scientific, Waltham, MA, USA) or p-nitrophenyl phosphate solution (Sigma-Aldrich) in tubes (30 min for the protein assay and 5 min for p-nitrophenyl phosphate in a dark room at room temperature). The conversion of p-nitrophenyl phosphate to p-nitrophenol was stopped using 3 N NaOH after 5 min. Absorbance (405 nm for p-nitrophenol analysis; 570 nm for protein analysis) was measured using a spectrophotometer (Bio-Rad Laboratories, Inc., Hercules, CA, USA). The ratio of ALP activity (calculated as p-nitrophenol value/WST-8 value) was calculated and used as “ALP index”. Previous reports [11] and our own data showed that the ALP index of bone-forming cells in the bone marrow was ≥1.0, and this value was used for quality assurance of cultured cells.

#### 2.6.2. Safety Tests

The culture medium with 10% autologous serum (1 mL) was subjected to three safety checks. To test for bacterial and fungal contamination, the medium was subjected to aerobic and anaerobic cultures using a membrane filter technique, which was performed at the last medium change before cell transplantation and at the time of cell transplantation.

The medium was tested for mycoplasma contamination using PCR, the mycoplasma-specific enzyme detection method, and the culture method. DNA was extracted from the prepared medium using phenol:chloroform:isoamyl alcohol (PCI) (Sigma-Aldrich). Equal volumes of PCI were added to the medium (600 µL) and centrifuged at 15,000 rpm for 5 min at room temperature. The supernatant of the centrifuged tube was mixed with 400 µL ice-cold 100% isopropanol (Wako Pure Chemical Industries, Ltd.) (400 µL) and centrifuged at 15,000 rpm for 10 min. The pellet was then rinsed with 400 µL ice-cold 70% ethanol (Wako, Osaka, Japan) and centrifuged at 15,000 rpm for 5 min. The final pellet was dried for 3 min and dissolved in distilled water (20 µL) for PCR. Two-step PCR was performed (Hasegawa et al., 1993). The reaction mixture for PCR contained 1 μL of each primer (10 pmol/μL), 5 μL of 10× reaction buffer, 10 nmol of each deoxynucleotide, samples as templates, and water to a volume of 49 μL. The primers used for the first-step PCR were MCGpF11(5′-ACACCATGGGAG(C/T)TGGTAAT-3′) and R23-JR(5′-CTCCTAGTGCCAAG(C/G)CAT(C/T)C-3′). The second-step PCR was also performed in a 50 µL volume with inner primers R16-2 (5′-CTG(C/G)FF(A/C)TGGATCACCTCCT-3′) and MCGpR21 (5′-GCATCCACCA(A/T)A(A/T)AC(C/T)CTT-3′). The reaction mixture contained 1 µL of the first PCR product. Thirty-five cycles were performed under the following conditions: denaturation at 94 °C for 30 s, annealing at 55 °C for 2 min, and polymerization at 72 °C for 2 min. Amplified DNA was separated on a 1.5% agarose gel and soaked in TAE buffer containing 0.1 µg/mL ethidium bromide. The products were analyzed using a ChemiDoc XRS gel documentation system (Bio-Rad Laboratories, Inc., Hercules, CA, USA). Mycoplasma contamination was also assessed using the MycoAlert^TM^ kit (LONZA KK., Tokyo, Japan). The ratio of luminescence before adding the substrate solution (measured value A) and luminescence 10 min after adding the substrate solution (measured value B) were measured and considered negative when the ratio was ≤1.0. In the mycoplasma culture method, the presence or absence of a mycoplasma colony was examined under a microscope at a magnification of 100 times or higher. If the results of the mycoplasma culture, PCR, and enzyme methods were all negative, it was considered mycoplasma-negative.

Endotoxin levels were measured at the time of the final medium exchange and at the time of cell transplantation. The Limulus ES-II Test Wako (299-51201, FUJIFILM Wako Pure Chemical Corporation, Osaka, Japan) was used, and the judgment was made with Toxinometer^®^ ET-201(FUJIFILM Wako Pure Chemical Corporation). Samples containing 1 EU/mL or less were considered negative.

#### 2.6.3. Flow Cytometric Analysis

The characteristics of the BMSCs were evaluated using flow cytometry in a previous study [7]. Aria flow cytometer (Becton Dickinson, San Jose, CA, USA) and the following antibodies were used: fluorescence isothiocyanate (FITC)-conjugated, phycoerythrin-conjugated, peridinin-chlorophyll-protein-conjugated (PerCP-Cy5.5), allophycocyanin-conjugated, Alexa Fluor 405-conjugated, or biotinylated antibodies against HLA-ABC, HLA-DR, CD3, CD14, CD19, CD34, CD73, CD90, CD106, CD146, CC chemokine receptor-5 (all from BD Pharmingen, Franklin Lakes, NJ, USA), CD10, CD29 (both from Dako, Glostrup, Denmark), and CD45 (Caltag; Invitrogen). Unconjugated CD105 antibody (Immunotech, Marseille Cedex 9, France) was covalently conjugated to FITC. The STRO-1 antibody (R&D Systems, Minneapolis, MN, USA) was detected using phycoerythrin-conjugated anti-mouse IgM. Biotinylated antibodies were detected using streptavidin Pacific Blue (Molecular Probes, Invitrogen, Carlsbad, CA, USA) or streptavidin PerCPCy5.5 (BD Pharmingen) conjugates. Propidium iodide (Dojindo) was used to detect dead cells. Color-conjugated mouse-IgG1k (BD Pharmingen) was used as a negative control. Data analysis was performed using the FlowJo software (TreeStar, San Carlos, CA, USA).

### 2.7. Preparation of Transplants

The cells in the scaffold were washed with saline three times and transferred to the operating room. The β-TCP granules were loosely connected to the cells and matrices and could be transferred as a mass in most cases.

### 2.8. Major Differences in Cell Culture and Processing Protocol between the Previous and Present Clinical Studies

In terms of the cell culture protocol, the results from an optimization study showed that the ability to form ectopic bone was immediately lost after passage [7]. Accordingly, BMSCs were cultured in flasks for just one passage in this study, then seeded onto the scaffold and induced into osteogenic cells (passage 2 cells), instead of the passage 3 cells used for osteogenic differentiation in a previous clinical study [6]. Because bFGF can enhance cell growth and help maintain the in vivo osteogenic ability, bFGF was added to the culture medium during the cell expansion phase [7]. Optimization of the mixing conditions for BMSCs and the scaffold was also considered. The results of our study showed that cell seeding onto scaffolds prior to osteogenic induction resulted in more new in vivo bone formation than cell seeding after osteogenic induction [12]. BMSCs were seeded onto the scaffolds before osteogenic induction.

### 2.9. Surgical Procedure

Transplantation was performed under local anesthesia and intravenous sedation with propofol. The lateral sinus wall was removed and the sinus floor mucous membrane (Schneiderian membrane) was carefully elevated. The transplant was filled in the space between the sinus floor and the elevated membrane. The sinus wall was repositioned and the mucoperiosteal flap was sutured. In cases of alveolar ridge augmentation, the mucoperitoneal flap was elevated, and the transplant was placed on the atrophic alveolar ridge where the dental implant installation was planned and covered with a GTR membrane or titanium mesh. After flap extension, the incision was sutured.

In both sinus floor elevation and alveolar ridge augmentation cases, dental implants were installed four months after the transplantation. Novel active implants (Nobel Biocare, Zürich-Flughafen Switzerland) were used for all cases.

Abutment connection was performed after 3 months for lower jaw implants and 6 months for upper jaw implants.

### 2.10. Evaluation

Panoramic X-rays were evaluated before the operation and at 6, 12, and 24 weeks and 1 and 2 years after the operation. Computed tomography (CT) was performed before the operation, at 12 and 24 weeks, and 1 and 2 years after the operation. The amount of regenerated bone was calculated using the SimPlant software (Materialize, Leven, Belgium). A bone biopsy was performed 16 weeks after cell transplantation at the time of dental implant installation using a trephine bur (2 mm inner diameter and 3 mm outer diameter; Stoma am Mark GmbH, Emmingen-Liptingen, Germany). After embedding in resin, non-decalcified ground sections were prepared, and the sections were evaluated using Villanueva Goldner staining and hematoxylin and eosin staining (H–E).

#### Histomorphometric Analysis

Light microscope images were captured using a digital camera (Carl Zeiss AG, Oberkochen, Germany) and transferred to a computer. The extent of new bone area, area of remaining scaffold, area of fibrous tissue, and area of bone marrow-like tissue were manually assessed using Image J [13] by an examiner who was well-trained in bone histology, was not a co-investigator, but was a member of this clinical study. The data were confirmed by a bone histology specialist. The size of these specific areas is expressed as a percentage of the total area of the section.

### 2.11. Statistical Analyses

For bone volume analysis, one-way ANOVA with Bonferroni correction was used for multiple comparisons as a post-test. A value of *p* < 0.05 was considered statistically significant. For the histomorphometric analysis of regenerated bone, we conducted a pre-specified Bayesian inferiority test using the method of Thall and Sung [14]. Specifically, if the posterior probability of the average bone area of current trial μ_C_ is larger than that of previous study μ_P_ minus non-inferiority margin δ (=13.92%), Pr[μ_C_ > μ_P_ – δ |y], is larger than 0.8, we consider our modified protocol was not inferior to the original protocol. Correlation coefficient was calculated using Excel (Microsoft Corporation, Redmond, Washington, DC, USA), and a correlation coefficient of less than 0.2 was considered as very weak or no association.

## 3. Results

### 3.1. Cases

Information regarding the study participants is summarized in Table 2. In this study, 15 participants (eight males and seven females) were enrolled and underwent bone marrow aspiration. Cell transplantation was performed in all patients with an average age of 51.3 years. Sinus floor elevation was performed at 20 sites in 11 cases, and alveolar ridge augmentation was performed at five sites in five cases; both procedures were performed in one case.

### 3.2. Characteristics of BMSCs Cultured with the Optimized Protocol

The results were originally reported in a previous study [7]. Briefly, BMSCs were positive for most of the examined MSC markers (CD10, CD29, CD73, CD90, CD105, and CD146), except for CD106 and STRO-1, and this did not change up to passage 3 (Appendix A (Figure A1a)). When bFGF was added to the culture medium, the expression profile of MSC markers was almost identical to that without bFGF, but the CD106- and STRO-1-positive fractions were slightly increased (Appendix A (Figure A1b)). Interestingly, the HLA-DR-positive/CD14- and chemokine receptor-5 (CCR5)-negative fractions in cells cultured with bFGF were larger than those in cells cultured without bFGF (Appendix A (Figure A1c,d)).

### 3.3. Cell Growth and Individual Variation

Primary cultured cells reached confluence between 11 and 25 days and showed a typical fibroblast-like morphology (Figure 1a). The average cell number at the time of cell harvest (P0) was 1.5 ± 0.33 × 10^7^ cells (Figure 1b), which was almost comparable to that of the preceding clinical study at P2 (1.6 × 10^7^) [6]. Although the number of harvested cells varied from 0.43 × 10^7^ to 2.1 × 10^7^, the cell proliferation rate (harvested cell number/seeded cell number) at P1 (Figure 1c) and cell proliferation speed (cell division number/day) at P1 (Figure 1d) were only slightly different, suggesting that the individual variation in the number of harvested cells was due to the difference in the initial cell number but not due to the individual cell characteristics. To support this, the harvested cell number became almost identical in the later cases when the technique for cell harvest and primary culture became more proficient.

### 3.4. Cell Differentiation

BMSCs from all the participants showed increased ALP activity after osteogenic induction (Figure 2a). The ALP index was higher than 1 in all cases (Figure 2b), fulfilling the minimal requirement for quality assurance of the BMSCs in this study.

### 3.5. Clinical Findings: Sinus Floor Augmentation

In cases of sinus floor augmentation, the lateral wall of the sinus was cut out, and the Schneiderian membrane was elevated (Figure 3a). The space was then filled with tissue-engineered bone (BMSCs cultured with β-TCP granules, as described above) (Figure 3b). All 11 patients showed uneventful healing, and no postoperative infections were observed.

Four months after cell transplantation, the mucoperiosteal flap was re-elevated (Figure 3c), and fixtures were installed (Figure 3d). An adequate volume of bone was generated in all cases, and dental implants were successfully installed. The hardness of the regenerated bone is almost identical to that of the surrounding existing bone.

The time course of the bone regeneration was observed by panoramic radiography. Representative panoramic radiographs are shown in Figure 4, which is the same case shown in Figure 3. The sinus floor augmentation cases lacked sufficient alveolar bone height in the upper molar regions, and the height was <5 mm (Figure 4a). The tissue-engineered bone was transplanted into the sinus floor (Figure 4b). After three months, the transplantation sites were evaluated by panoramic radiography before implant installation (Figure 4c). After 6 months, the augmented sinus floor became flat, and the shape of the transplanted β-TCP granules was almost invisible at this stage (Figure 4d). At 6–12 months, the morphology of the regenerated bone became close to that of the surrounding existing bone and matured with clear separation of cortical and cancellous bone (Figure 4e), which was well-maintained for up to 24 months (Figure 4f).

### 3.6. Clinical Findings: Alveolar Ridge Augmentation

In cases of alveolar ridge augmentation, the mucoperiosteal flap was elevated, and the atrophic alveolar bone was exposed (Figure 5a). The tissue-engineered bone was placed onto the atrophic bone, covered with a membrane (Figure 5b), and then sutured. All five patients showed uneventful healing, and no postoperative infection was observed.

After 4 months, the mucoperiosteal flap was re-elevated (Figure 5c). The surface of the regenerated bone was covered with relatively hard bone, but β-TCP granules were still visible. Drilling was performed, and dental implants were installed in all cases, although the hardness of the regenerated bone varied among individuals.

### 3.7. CT Images

The CT images of representative cases are shown in Figure 6. Sinus floor augmentation was performed in cases with bone resorption at upper molar region (Figure 6a). In the cases of sinus floor augmentation, the transplanted tissue-engineered bone initially showed an irregular margin, and the β-TCP granules were still recognizable at 3 months (Figure 6b). After 6 months, β-TCP granules were mostly degraded, but the trabecular structure was not regenerated (Figure 6c). After 12 months, the trabecular structure gradually formed, and the border between the regenerated and original existing bone became unclear (Figure 6d). At 24 months after transplantation, the trabecular structure was mostly regenerated, and the regenerated bone was hard to distinguish from the surrounding existing bone (Figure 6e). Alveolar ridge augmentation with tissue-engineered bone was performed in cases with narrow alveolar ridges (Figure 6f). At 3 months, the border between the transplanted tissue-engineered bone and the surrounding existing bone was clear and β-TCP granules remained, as shown in the sinus floor augmentation cases (Figure 6g). At 24 months after transplantation, the border between the regenerated bone and existing bone became unclear (Figure 6h).

### 3.8. Volume Change of the Regenerating Bone

The volume of regenerated bone was analyzed using a software and the CT images at 3, 6, 12, and 24 months after cell transplantation. The percentage of bone volume at each time point compared to that at 3 months was calculated (Figure 7). In general, the volume decreased over time, although there were some cases in which the volume was well-maintained for up to 24 months (data not shown). The average volume of regenerated bone at 6 months was 99% of that at 3 months, which dropped immediately after 6 months, 80% at 12 months, and 63% at 12 months (Figure 7).

### 3.9. Histomorphometric Analyses

A bone biopsy was performed at 4 months at the site of implant installation using a trephine bur. The histology of a representative case is shown in Figure 8a. Non-decalcified sections were stained using the Villanueva–Goldner staining. The green area shows mature bone and the red area shows immature bone. The remaining β-TCP granules were observed in some areas and are rendered in gray color. The remainder of the specimen was filled with fibrous connective tissue. The average new bone area was 41.51%, and β-TCP granules occupied 10.61% (Figure 8b). Fibrous connective tissue occupied 47.88%. Bone-marrow-like tissues were not observed in the present study.

### 3.10. Comparison of New Bone Area in the Present Study and the Preceding Clinical Study

The percentage of the new bone area was the endpoint of the previous and present clinical studies. Since the purpose of this study was to investigate the effect of different cell-processing protocols on clinical efficacy and compare the results with those of a previous clinical study, the percentage of new bone area was compared using a box-and-whisker plot (Figure 9). In both studies, the average area of new bone was approximately 42%. In contrast, the large individual variation shown in the preceding study was dramatically improved in the present study, and only limited variation was observed. The histological analysis was performed 4 months after transplantation in this study, which was 2 months earlier than that in the previous study (6 months after transplantation).

### 3.11. Correlation Analyses between the New Bone Area and Cellular Parameters

It was our interest to determine if these positive changes were related to some cellular parameters. Accordingly, the correlations between the new bone area and various parameters, including harvested cell number (Figure 10a), cell proliferation ratio (Figure 10b), cell growth speed (Figure 10c), ALP activity (Figure 10d), and ALP index (Figure 10e), were analyzed. However, none of these parameters showed a significant correlation.

### 3.12. Dental Implants

In total, 56 dental implants were installed in the regenerated bone, all of which were integrated (100%). During the follow-up period, all implants were maintained without difficulty for 24 months.

### 3.13. Adverse Effects and Safety Issue

No side effects related to cell transplantation were observed during the treatment or follow-up period.

## 4. Discussion

In this study, an optimized protocol for cell processing was used [7,12]. The optimized protocol limited the passage number, and bFGF was added to the culture medium, which may have affected the quality of the cells. A sufficient number of cells were harvested at passage 1 in all cases, and the cell number was identical to that of the previous study at passage 2 (1.5 × 10^7^ cell versus 1.6 × 10^7^ cells, respectively). In the preceding study, there was a case for which the protocol was discontinued due to the lack of cell proliferation. In this study, a sufficient number of cells was obtained in all cases (15 out of 15 cases). Although the results from only one case cannot be generalized, it is conceivable that the increased volume of bone marrow aspirate (20 mL vs. 10 mL) and the addition of bFGF in the culture medium contributed to the stable cell growth to sufficient number [7]. In patients Nos. 1 and 2, the total cell number was relatively smaller than that in the other patients, although the cell proliferation rate (population doubling) was almost identical. This suggests that the difference might be due to the initial number of BMSCs in the bone marrow aspirate, but not due to the proliferation capability of BMSCs. Despite stable cell proliferation, ALP assay results showed significant individual variation. Although ALP is a conventional marker for osteogenic differentiation and has been widely used in both basic and clinical studies [6], our previous study showed that the level of ALP activity did not parallel the in vivo osteogenic capability. The results of this study confirmed this finding, suggesting the limited usefulness of ALP activity for the quality assurance of BMSCs.

Safety was the primary endpoint of this clinical study. There were no side effects related to cell transplantation during the observation period (24 months) and up to now (8–10 years), which was comparable to the previous clinical study with the original protocol (no side effects or health concerns were noted) [6]. The major safety concerns for tissue engineering using BMSCs include possible infection from transplanted cells and tumorigenic transformation. In this study, autologous cells were used to avoid potential donor infection. For the culture of individual cells, safety tests were performed, including the endotoxin test, mycoplasma test, and bacteriological examination. These commonly used tests appear to be effective in supporting the safety of cell transplantation. Tumorigenic transformation is considered a potential risk factor for BMSCs [9]. In particular, genetic instability in BMSCs has been reported, which could cause malignant transformation [15]. However, this is unlikely, and no such complications have been reported so far [4,5,6,16,17]. The results from the present study further support the safety of BMSCs.

Fifteen participants were enrolled in this study, all of whom showed successful bone regeneration, which was sufficient to support implant instillation. Histomorphometric analysis of the bone biopsy sample was performed as the secondary endpoint. We conducted pre-specified Bayesian inferiority test using the method of Thall and Sung [14]. The results showed that the average bone area was 41.51% in this trial, which was comparable to that in our previous study (41.9%) and Pr[μ_C_ > μ_P_ − δ |y] was 0.884; thus, the non-inferiority of this protocol was proven (*p* < 0.05). Importantly, bone biopsy was performed 2 months earlier in the present study (4 months vs. 6 months after transplantation). Osteointegration of the implants installed in the tissue-engineered bone was 100%, and no implants were lost during the observation period, confirming the successful regeneration of functional bone.

It is well-known that regenerated bone shows resorption, which was also observed in tissue-engineered bone in our previous study [6]. In this study, the volume of the regenerated bone was reduced to 63% over 24 months. This is one of the drawbacks of this protocol and should be considered when planning an operation. The histology of regenerated bone showed significant degradation and absorption of the β-TCP scaffold, which should be the reason for the initial volume reduction. However, radiographic images showed that the scaffold degraded after 12 months, and the morphology of the regenerated bone appeared to be like-normal by this stage, suggesting that the absorption after 12 months might be due to bone remodeling.

One of the major limitations of this study was the usage of a non-inferiority trial. A randomized controlled trial that compares this method with a clinical gold standard such as the autologous bone graft would be a preferable approach for future studies. In this study, autologous cells were used, which can avoid the triggering of immunological reactions and are safe in nature. However, their cell processing for clinical usage requires a GMP-compliant facility, and the process is highly labor-intensive. For the widespread usage of bone tissue engineering, allogenic cells should be considered; the development of a cell product such as an advanced therapy medicinal products (ATMP) is mandatory.

The optimization of the cell-processing protocol in this study was performed based on the in vivo bone-forming capability of cells, as the known osteogenic marker expression was not necessarily parallel to the level of bone formation in vivo [7]. The results of this study clearly showed that BMSCs processed with the optimized protocol demonstrated bone regeneration with less individual variation compared to BMSCs processed with the conventional protocol. This suggests that an ectopic transplantation animal model using human cells can be used as a reliable model to predict clinical efficacy, which is important for future clinical studies. In contrast, as shown in an in vivo animal study [7], all cell characteristic markers used in this study, such as cell proliferation and cell differentiation markers, did not correlate with clinical efficacy. The expression of higher levels of osteogenic markers may confirm the presence of osteogenic cells but does not support osteogenic function in vivo. This is a major drawback for quality assurance of cultured cells for bone tissue engineering, and further studies are needed.

## 5. Conclusions

The results from this clinical study showed that the optimized cell-processing protocol contributed to faster bone regeneration with less individual variation. The ectopic transplantation model of human cells into immunodeficient animals can be used for optimization of the cell-processing protocol, since the results were comparable to the efficacy of clinical studies in bone tissue engineering. However, the quest for functional markers of BMSCs remains unresolved and should be the focus of future research.

## Figures and Tables

**Figure 1 jcm-11-07328-f001:**
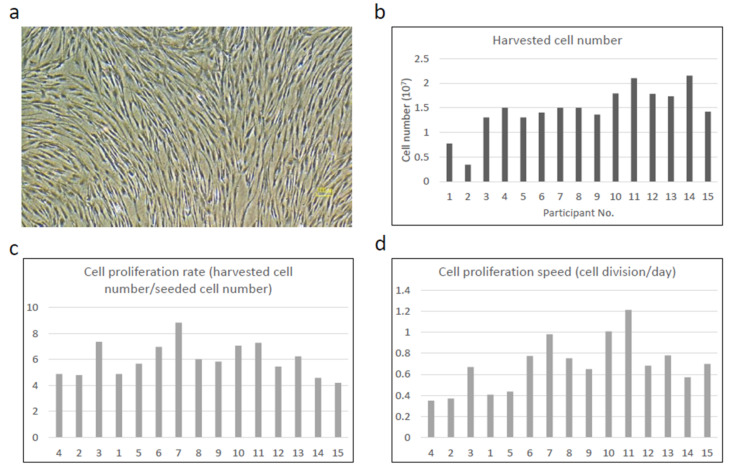
Cell growth of BMSCs. Representative phase contrast photomicrograph of the cells at the time of cell harvest: (**a**) harvested cell number (P0); (**b**) cell proliferation rate (harvested cell number/seeded cell number) (P1); (**c**) and cell growth speed (cell division number/day) (P1). (**d**) Although the number of harvested cells was low in No. 1 and 2, the cell proliferation rate and cell proliferation speed of those patients were only slightly different. The individual variation of those parameters was within the acceptable level in all patients.

**Figure 2 jcm-11-07328-f002:**
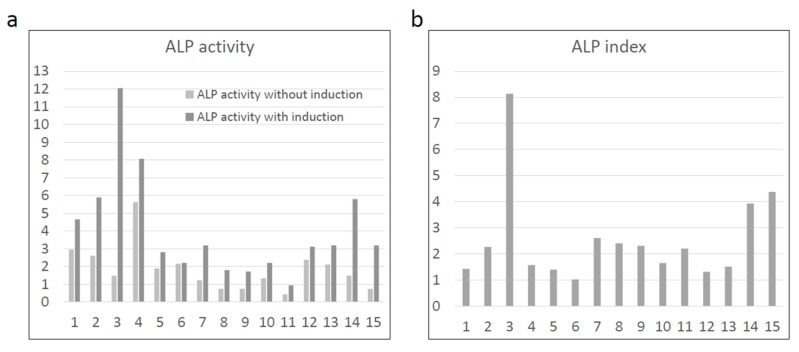
Osteogenic differentiation capability of BMSCs. (**a**) Alkaline phosphatase (ALP) activity were increased in all samples after osteogenic induction. (**b**) All samples showed ALP index higher than 1, fulfilling the requirement of quality assurance in this study.

**Figure 3 jcm-11-07328-f003:**
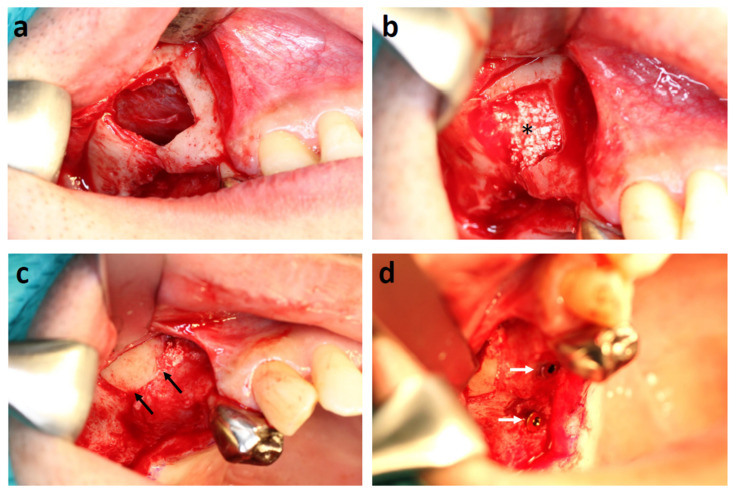
Representative case of maxillary sinus floor augmentation. (**a**) Elevation of the Schneiderian membrane. (**b**) Tissue-engineered bone filled into the space between the alveolar bone and Schneiderian membrane (asterisk indicates tissue-engineered bone). (**c**) Regenerated alveolar bone at 16 weeks after transplantation (black arrows indicate transplanted site). (**d**) Dental implant installation (white arrows).

**Figure 4 jcm-11-07328-f004:**
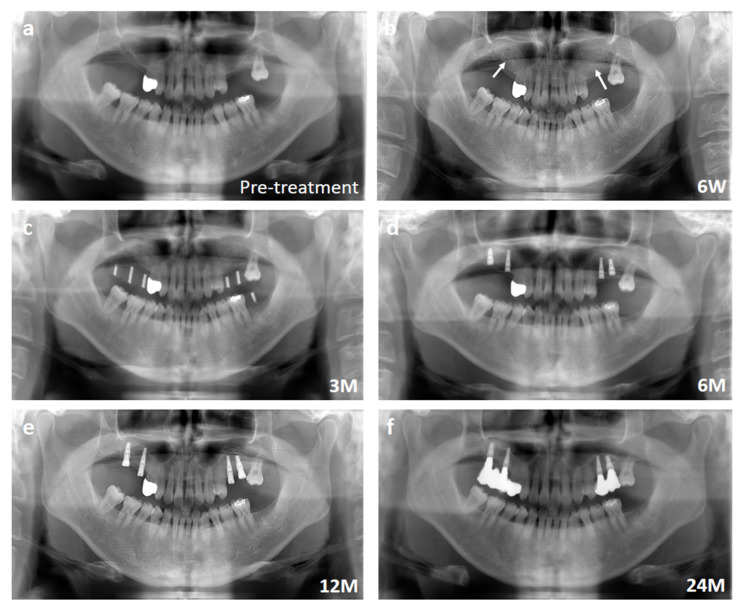
Representative panoramic radiographs of the sinus floor augmentation case. (**a**) Pre-treatment. Note that the height of alveolar bone was extremely thin. (**b**) Six weeks after transplantation. White arrows indicate the elevated sinuses. (**c**) Three months after transplantation. (**d**) Six months after transplantation. (**e**) Twelve months after transplantation. (**f**) Twenty-four months after transplantation.

**Figure 5 jcm-11-07328-f005:**
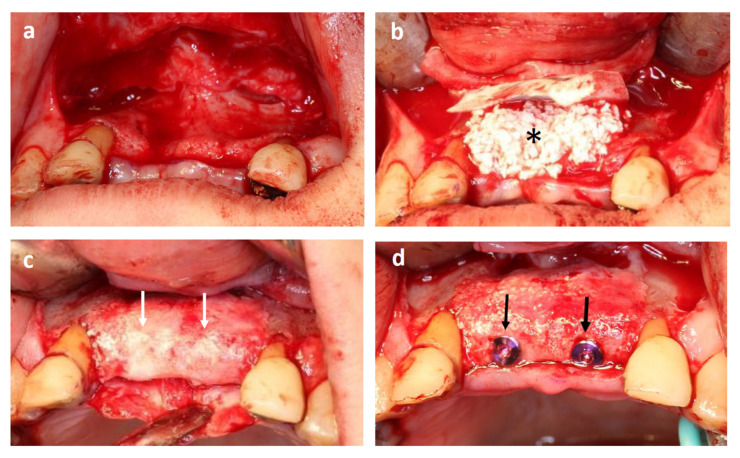
A representative case of alveolar ridge augmentation. (**a**) Elevated mucoperiosteal flap. Note the thin alveolar bone. (**b**) Tissue-engineered bone put onto the buccal side of the alveolar bone. (* indicates the tissue-engineered bone). (**c**) Four months after transplantation (white arrows indicate regenerated bone). (**d**) Dental implants were installed into the regenerated bone (black arrows).

**Figure 6 jcm-11-07328-f006:**
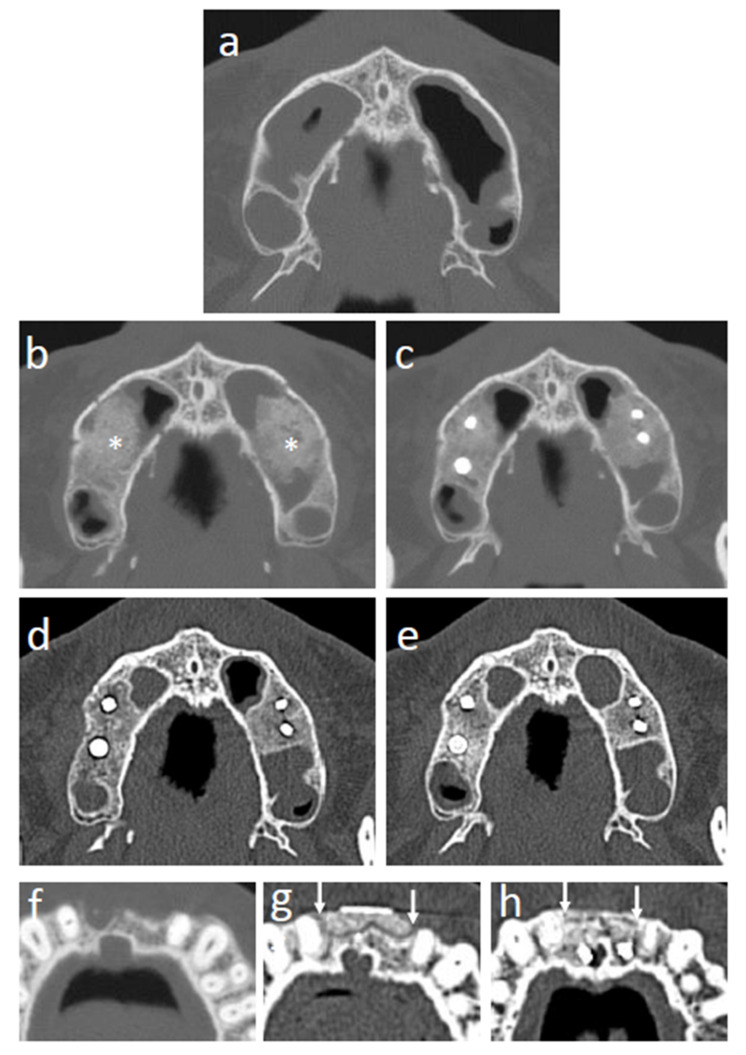
CT images of regenerating sites. (**a**) Sinus floor augmentation case before transplantation. (**b**) Sinus floor augmentation case at 3 months after transplantation (* shows the transplanted tissue-engineered bone). (**c**) Sinus floor augmentation case at 6 months after transplantation. (**d**) Sinus floor augmentation case at 12 months after transplantation. (**e**) Sinus floor augmentation case at 24 months after transplantation. (**f**) Alveolar ridge augmentation case before transplantation. (**g**) Alveolar ridge augmentation case at 3 months after transplantation (white arrows indicate the area of transplanted tissue-engineered bone). (**h**) Alveolar ridge augmentation case at 24 months after transplantation (white arrows indicate the area of transplanted tissue-engineered bone).

**Figure 7 jcm-11-07328-f007:**
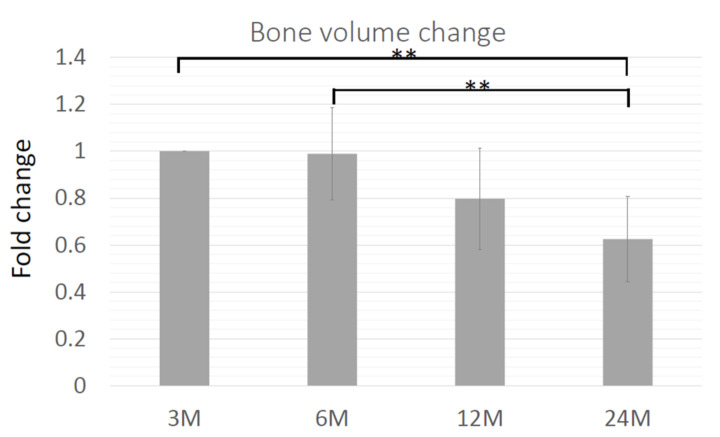
Time course of bone volume. The relative volume of the regenerating bone was shown as the percentage of volume at 3 months. ** *p* < 0.01.

**Figure 8 jcm-11-07328-f008:**
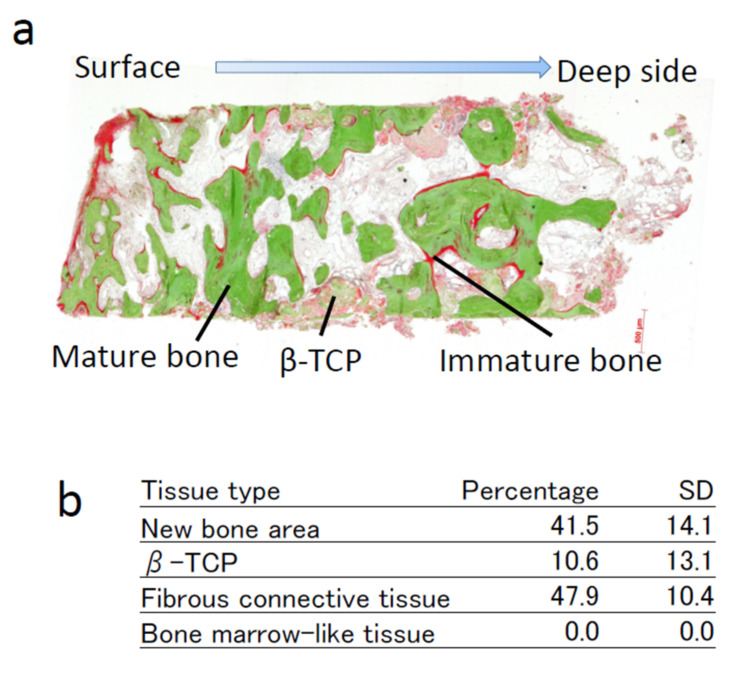
Histomorphometric analysis of bone biopsy samples at 4 months after transplantation. (**a**) Photomicrograph of a representative tissue section. Villanueva–Goldner staining of a non-decalcified section. (**b**) Average percentages and standard deviations (SD) of each tissue in bone biopsy samples.

**Figure 9 jcm-11-07328-f009:**
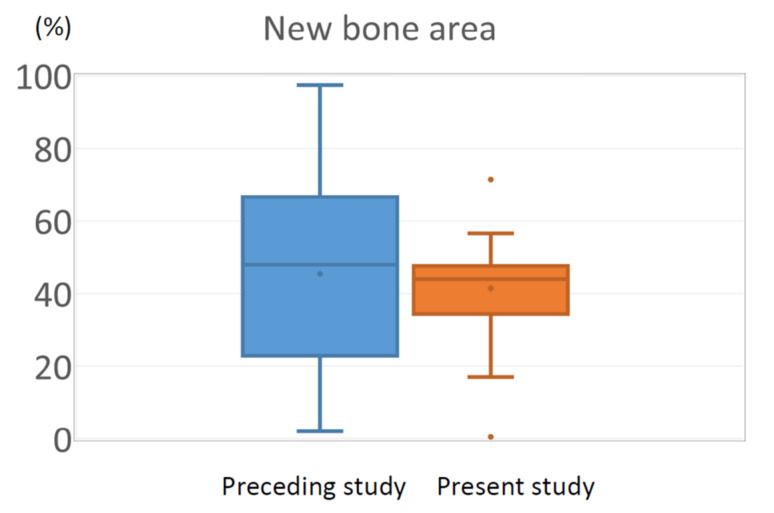
Box-and-whisker plot showing the distribution of new bone area in the previous and present studies. Note the variation was considerably improved in the present study.

**Figure 10 jcm-11-07328-f010:**
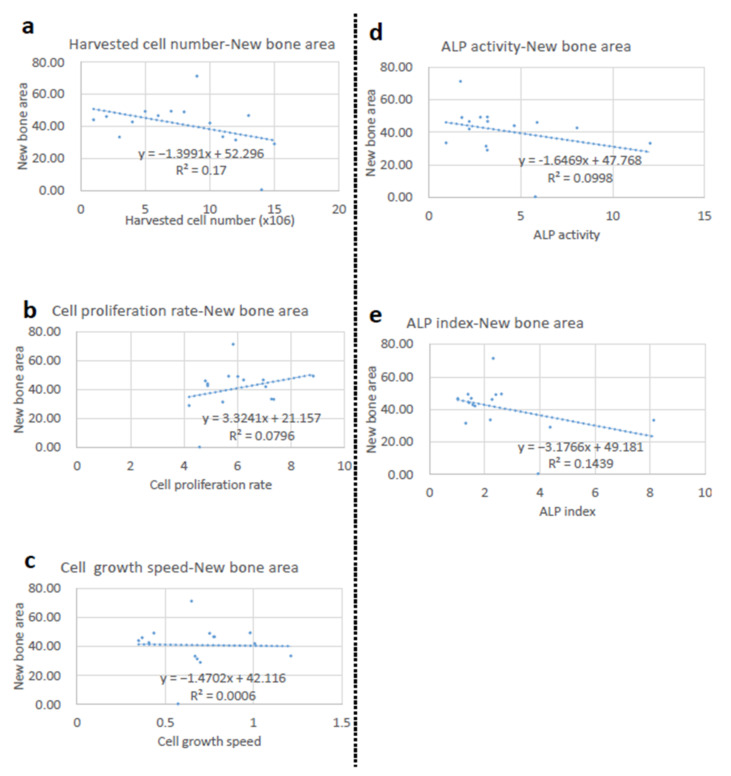
Correlation between the new bone area and cellular parameters. (**a**) Correlation between harvested cell number and new bone area. (**b**) Correlation between cell proliferation rate and new bone area. (**c**) Correlation between cell growth speed and new bone area. (**d**) Correlation between ALP activity and new bone area. (**e**) Correlation between ALP index and new bone area.

**Table 1 jcm-11-07328-t001:** Study design and schedule for the intervention.

Intervention	Pre-Treatment	Day 0	6 W	3 M	4 M	6 M	12 M	24 M
Verification of the selection criteria, and informed consent obtained	✓							
Blood pressure and pulse rate measurement	✓	✓			✓			
Objective symptom, oral examination	✓		✓	✓	✓	✓	✓	✓
Blood test	✓				✓			
Urinalysis	✓							
Chest X-ray	✓							
ECG	✓							
Panoramic X-ray	✓		✓					
CT	✓							
Bone marrow aspiration, blood collection	✓							
Cell transplantation								
Bone biopsy, implant installation								

W, weeks; M, months; CT, computed tomography; ECG, electrocardiogram.

**Table 2 jcm-11-07328-t002:** Summary of patient information.

No.	Age	Sex	Target Region and Procedure	Number of Implants
1	67	F	Lt. SFA + U.ARA	2
2	64	M	Lt. SFA	2
3	43	F	Bil. SFA	5
4	63	M	Bil. SFA	4
5	60	F	Bil. SFA	6
6	44	F	Bil. SFA	5
7	64	M	Bil. SFA	4
8	38	M	Bil. SFA	4
9	23	F	L. ARA	2
10	55	M	Bil. SFA	6
11	53	M	Bil. SFA	5
12	57	M	Bil. SFA	5
13	50	F	U. ARA	2
14	50	M	U. ARA	2
15	38	F	U. ARA	2

Fifteen patients participated in this study. Sinus floor augmentation (SFA) and/or alveolar ridge augmentation (ARA) were performed in 11 and 5 patients, respectively. Rt, right side; Lt, left side; Bil, bilateral sides; U, upper jaw; L, lower jaw.

## Data Availability

Detailed information about the clinical study can be obtained from a public database: “UMIN Clinical Trials Registry (UMIN000006255)” at https://center6.umin.ac.jp/cgi-open-bin/ctr/ctr_view.cgi?recptno=R000007387 (accessed on 28 September 2022).

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
