# Peer review of "A Clinical Study of Alveolar Bone Tissue Engineering Using Autologous Bone Marrow Stromal Cells: Effect of Optimized Cell-Processing Protocol on Efficacy"

_jcm, 2022, doi:10.3390/jcm11247328_

Round 1

Reviewer 1 Report

In this original article, the priming of a cell-processing protocol on the clinical efficacy of bone tissue 24 engineering in preparation for dental implantation was investigated. The authors provide a detailed, well-founded and linguistically precise description of the design and implementation of the study. The presentation of the results is clear and the evaluation is reflective. The study will be a valuable contribution to the further development of tissue engineering techniques in dentistry.

Dear authors, a few small comments could be a useful addition, especially for colleagues with a clinical focus:

How many patients were excluded from the study? Was there any drop-out?

How long was the healing time of the implants? Did the patients wear dentures during the healing period?

Which implants were placed and which surgical protocol was chosen for implant placement? For example, was additional bone expansion required in the maxilla?

Author Response

I would like to thank the reviewer for the positive and useful comments.

  1. How many patients were excluded from the study? Was there any drop-out?

-The participants were referred from other hospitals/clinics and primarily fulfilled the eligibility. No one was excluded after examination and there was no drop out. This comment was added to the Materials and method section.

  1. How long was the healing time of the implants? Did the patients wear dentures during the healing period?

-Abutment connection was performed after 3 months for lower jaw implants and 6 months for upper jaw implants. This was also added to the Materials and method section.

  1. Which implants were placed and which surgical protocol was chosen for implant placement? For example, was additional bone expansion required in the maxilla?

-The type and makes of the implants used in this study, and the surgical protocol were added in Materials and methods section. No additional bone expansion was performed in this study.

Reviewer 2 Report

This manuscript describes that an optimized cell processing protocol by the limited cell passage number and the bFGF addition into the culture medium leads to the bone regeneration with comparable to the preceding protocol and this bone regeneration is rapid and stable. This study is relatively well designed and conducted.  However, this manuscript needs to be addressed to some points as described below.

General comments:

There is no description about the statistical analysis method although the results by correlation analysis and shown in Figure 10 and the sentence regarding the statistical analysis was described in Lines 473-479. Please describe about the statistical analysis methods in “Materials and Methods” section.

Line 254: Please describe about the examiner in more detail. Is an examiner a specialist on histology and not related to this clinical study?

Please describe the reason why the bone volume decreased over time up to 24 months. As one reason, is it conceivable that the degradation or adsorption of b-TCP was caused?

Regarding Figures 7 and 8, are the differences of bone volume and histological changes found between SFA and ARA procedures?

Supplementary Figure 1b shows that the expression level of CD106 in the presence of bFGF was slight positive. Were individual differences in this expression level found? Was this increased expression level also significantly different with compared to the culture condition without bFGF?

Supplementary Figure 1d: Please present the histogram in the case of the culture condition without bFGF.

Specific comments:

Line 109: Please delete “f” just after “cell transplantation.”.

Line 143: Please delete “m” just after “with 400”.

Line 177: Please delete “m” between “1” and “ml”.

Line 241: Please change “(Fig. 1a)” to “(Table 1)”.

Lines 258 and 264: Please change “Table 1” to “Table 2”.

Line 369: Please change “(Fig. 6f)” to “(Fig. 6g)”.

Line 374: Please change “12 months” to “24 months”.

Line 447: Please change “numbe” to “number”.

Figure 1b: “106” on Y-axis should be changed to “107”.

Figure 6: A CT image at pre-treatment (before transplantation) should be added.

Figure 7: Please present the Y-axis or create as a table instead of figure graph. In addition, please show the result by statistical analysis.

Figure 8b: Please create as a table with SD values instead of figure graph.

Author Response

Thank you for your detailed comments and useful suggestions.

  1. There is no description about the statistical analysis method although the results by correlation analysis and shown in Figure 10 and the sentence regarding the statistical analysis was described in Lines 473-479. Please describe about the statistical analysis methods in “Materials and Methods” section.

-The statistical methods were described in Materials and Methods section as suggested.

  1. Line 254: Please describe about the examiner in more detail. Is an examiner a specialist on histology and not related to this clinical study?

-The explanation about the examiner was added.

  1. Please describe the reason why the bone volume decreased over time up to 24 months. As one reason, is it conceivable that the degradation or adsorption of b-TCP was caused?

-The histology of regenerated bone showed significant degradation and absorption of b-TCP scaffold and this should be the reason for initial volume reduction. However, radiographic images showed that the scaffold degraded after 12 months and the morphology of the regenerated bone seems like almost normal by this stage, suggesting that the absorption after 12 months might be due to bone remodeling. This comment was added to Discussion section.

  1. Regarding Figures 7 and 8, are the differences of bone volume and histological changes found between SFA and ARA procedures?

-Thank you for this important comment. I have compared the bone volume between SFA and ARA cases. However, possibly due to the limited number for ARA cases, no clear difference was noted. Similarly, the difference in histology between SFA and ARA was not observed.

  1. Supplementary Figure 1b shows that the expression level of CD106 in the presence of bFGF was slight positive. Were individual differences in this expression level found? Was this increased expression level also significantly different with compared to the culture condition without bFGF?

-Although there is some individual variation, it is my understanding that CD106 was slightly positive in MSC with bFGF, while that was almost negative in MSC without bFGF (see histograms in the attached file). This might be related to higher stemness of MSC with bFGF but it was beyond the scope of this study and not confirmed yet.

  1. Supplementary Figure 1d: Please present the histogram in the case of the culture condition without bFGF.

-Unfortunately, the data we have for BM-MSC without bFGF was from different cell source and performed for different experiment. So it may not appropriate to compare.  I just show the results here (please find the histogram in the attached file) for your reference. I hope this is acceptable.

  1. Specific comments:-I appreciate these detailed comments. They are very helpful.

Line 109: Please delete “f” just after “cell transplantation.”.-Deleted.

Line 143: Please delete “m” just after “with 400”.-Deleted.

Line 177: Please delete “m” between “1” and “ml”.-Deleted.

Line 241: Please change “(Fig. 1a)” to “(Table 1)”.-I am sorry about this confusion. It was deleted.

Lines 258 and 264: Please change “Table 1” to “Table 2”.-I deleted “table 2”.

Line 369: Please change “(Fig. 6f)” to “(Fig. 6g)”.-Changed.

Line 374: Please change “12 months” to “24 months”.-Changed.

Line 447: Please change “numbe” to “number”.-Changed

Figure 1b: “106” on Y-axis should be changed to “107”.-Changed.

Figure 6: A CT image at pre-treatment (before transplantation) should be added.-The image was added.

Figure 7: Please present the Y-axis or create as a table instead of figure graph. In addition, please show the result by statistical analysis.-The graph changed and the results from statistical analysis was also added.

Figure 8b: Please create as a table with SD values instead of figure graph.-A table was created.

Reviewer 3 Report

In the present manuscript, the authors describe a clinical study using bone marrow stromal cells in combination with beta-tricalcium phosphate as a tissue engineering approach for use in alveolar bone ridge and sinus augmentation. In it, they describe an optimized manufacturing process of their product and compare the results with a historical cohort from a previously published study. In principle, the work is quite interesting and scientifically sound, so the results should warrant publication. However, I feel that the conclusions are not supported by the available data. The authors state that the optimized protocol results in comparable bone quality and in a shorter regeneration time with less deviation. The non-inferiority for the procedure would thus be proven. At the same time, however, they state that none of the parameters studied show a positive correlation with bone quantity. In this respect, the question arises as to what influence unlisted parameters have. The anatomical conditions of the surgical sites (residual ridge height, sinus widths) might have influenced the histomorphometric results. The incomplete information available on these and other potentially confounding factors poses the risk of intransitivity. In addition, the study lacks any statistical analysis that would be necessary to support such a statement (hypothesis, delta…). However, it is fundamentally questionable whether the non-inferiority is relevant at all. In that case, the authors would have to clearly demonstrate that the described treatment is superior to a placebo/control. This has not been done in any study so far and therefore this protocol is not clinically relevant either. In addition, it would of course be nice to have not only a placebo but a clinical gold standard as a control. For example, what is the advantage of the very elaborate methodology that requires a GMP-compliant environment and validated methods to an approach that only uses an autologous cell concentrate? It is not clear whether the described method, which also uses as starting material an inhomogeneous cell population, selected only by plastic adherence, offers any advantage at all. If one sets the immense effort for the production of such an ATMP against it, also taking into account the regulatory requirements, the outcome should, in my opinion, be significantly better than current clinically established procedures in order to be able to gain clinical acceptance at all. These points should also be at least taken up in the discussion.

Ultimately, the optimized procedure leads neither to significantly more bone nor to faster regeneration (the comparison of two different examination time points 4 months vs. 6 months from different studies is not permissible).  The study thus merely describes a simplified manufacturing process in a clinical proof-of-concept and should be described as such.

Author Response

I appreciate the following important comments of this reviewer.

“In the present manuscript, the authors describe a clinical study using bone marrow stromal cells in combination with beta-tricalcium phosphate as a tissue engineering approach for use in alveolar bone ridge and sinus augmentation. In it, they describe an optimized manufacturing process of their product and compare the results with a historical cohort from a previously published study. In principle, the work is quite interesting and scientifically sound, so the results should warrant publication. However, I feel that the conclusions are not supported by the available data. The authors state that the optimized protocol results in comparable bone quality and in a shorter regeneration time with less deviation. The non-inferiority for the procedure would thus be proven. At the same time, however, they state that none of the parameters studied show a positive correlation with bone quantity. In this respect, the question arises as to what influence unlisted parameters have. The anatomical conditions of the surgical sites (residual ridge height, sinus widths) might have influenced the histomorphometric results. The incomplete information available on these and other potentially confounding factors poses the risk of intransitivity.”

Response: I understand this reviewer’s anxiety about the data interpretation. The major purpose of this study was to test the efficacy of optimized protocol under clinical circumstance. Importantly, this optimization was based on the results from in vivo animal study but not the parameters listed in this study. Our previous study (Agata et al., Tissue Eng Part A, 2010) showed that the in vivo bone forming capability was not parallel to the known osteogenic parameters such as ALP activity, so it was my interest to teste whether this was also true under clinical circumstances or not. I hope this explanation is understandable. However, I agree with this reviewers comment that the bone regeneration capability of cells could be affected by unknown/unlisted parameters, which might be able to use for the quality assurance of cells for bone tissue engineering. This should be investigated in future studies.

“In addition, the study lacks any statistical analysis that would be necessary to support such a statement (hypothesis, delta…). “

Response: Thank you for this suggestion. I have added statistical analyses for the results when necessary.

“However, it is fundamentally questionable whether the non-inferiority is relevant at all. In that case, the authors would have to clearly demonstrate that the described treatment is superior to a placebo/control. This has not been done in any study so far and therefore this protocol is not clinically relevant either. In addition, it would of course be nice to have not only a placebo but a clinical gold standard as a control. For example, what is the advantage of the very elaborate methodology that requires a GMP-compliant environment and validated methods to an approach that only uses an autologous cell concentrate? It is not clear whether the described method, which also uses as starting material an inhomogeneous cell population, selected only by plastic adherence, offers any advantage at all. If one sets the immense effort for the production of such an ATMP against it, also taking into account the regulatory requirements, the outcome should, in my opinion, be significantly better than current clinically established procedures in order to be able to gain clinical acceptance at all. These points should also be at least taken up in the discussion.”

Response: I agree with this reviewer’s comments and the limitation of this study was added in Discussion section as follows:

“One of the major limitations of this study was the usage of non-inferiority trial. A randomized controlled trial comparing with a clinical gold standard such as autologous bone graft should be favorable in the future studies. In this study, autologous cells were used, which can avoid immunological reactions and safety in nature. However, the cell processing for clinical usage requires a GMP-compliant facility and the process is highly labor intensive. For the widespread usage of bone tissue engineering, the possibility of allogenic cells should be considered and the development of a cell product such as an advanced therapy medicinal products(ATMP) is mandatory.”

“Ultimately, the optimized procedure leads neither to significantly more bone nor to faster regeneration (the comparison of two different examination time points 4 months vs. 6 months from different studies is not permissible).  The study thus merely describes a simplified manufacturing process in a clinical proof-of-concept and should be described as such.”

Response: I agree with this comment and the manuscript has been revised accordingly.